# A Review of Bullous Dermatologic Adverse Events Associated with Anti-Cancer Therapy

**DOI:** 10.3390/biomedicines11020323

**Published:** 2023-01-24

**Authors:** Rose Parisi, Hemali Shah, Neil H. Shear, Michael Ziv, Alina Markova, Roni P. Dodiuk-Gad

**Affiliations:** 1Albany Medical College, Albany, NY 12208, USA; 2Division of Dermatology, Department of Medicine, University of Toronto, Toronto, ON M5S 1A1, Canada; 3Department of Dermatology, Emek Medical Center, Afula 1834111, Israel; 4Dermatology Service, Department of Medicine, Memorial Sloan Kettering Cancer Center, New York, NY 10021, USA; 5Weill Cornell Medical College, Cornell University, New York, NY 10021, USA; 6Department of Dermatology, Bruce Rappaport Faculty of Medicine, Technion Institute of Technology, Haifa 3525433, Israel

**Keywords:** bullous, dermatologic adverse events, cutaneous adverse events, anti-cancer therapy, immunotherapy, immune checkpoint inhibitors, chemotherapy, targeted therapy

## Abstract

The rapid evolution of anti-cancer therapy (including chemotherapy, targeted therapy, and immunotherapy) in recent years has led to a more favorable efficacy and safety profile for a growing cancer population, and the improvement of overall survival and reduction of morbidity for many cancers. Anti-cancer therapy improves outcomes for cancer patients; however, many classes of anti-cancer therapy have been implicated in the induction of bullous dermatologic adverse events (DAE), leading to reduced patient quality of life and in some cases discontinuation of life-prolonging or palliative therapy. Timely and effective management of adverse events is critical for reducing treatment interruptions and preserving an anti-tumor effect. Bullous DAE may be limited to the skin or have systemic involvement with greater risk of morbidity and mortality. We present the epidemiology, diagnosis, pathogenesis, and management of bullous DAE secondary to anti-cancer therapies to enable clinicians to optimize management for these patients.

## 1. Introduction

The rapid evolution of anti-cancer therapy (including chemotherapy, targeted therapy, and immunotherapy) in recent years has led to a more favorable efficacy and safety profile for a growing cancer population and improved overall survival and reduced morbidity for many cancers. Enhanced anti-cancer therapy tolerance allows more patients to stay on treatment for longer durations leading to higher anti-cancer therapy utilization and an increased incidence and prevalence of associated adverse events (AEs) [1]. Timely and effective management of AEs is critical for reducing treatment interruptions and preserving an anti-tumor effect. Dermatologic AEs (DAEs) make up to 30–50% of treatment-associated AEs, with 1–5% being bullous DAEs [2,3]. Bullous DAEs consist of vesiculobullous eczema, hand–foot skin reaction, toxic erythema of chemotherapy, bullous pemphigoid, bullous lichenoid drug eruption, lichen planus pemphigoides, pemphigus vulgaris, bullous erythema multiforme, linear IgA bullous dermatosis, bullous lupus erythematosus, Stevens–Johnson syndrome (SJS)/toxic epidermal necrolysis and SJS-like eruptions, and non-specific bullous drug eruption [4,5,6,7,8,9]. 

Chemotherapy is defined as the use of cytotoxic chemicals to destroy rapidly growing and differentiating cells. Chemotherapeutic drugs can be distinguished into a number of classes, including anti-metabolites, anthracyclines, alkylating agents, anti-microtubular agents, methylation inhibitors, topoisomerase inhibitors, and vinca alkaloids. These are the oldest and most established form of anti-cancer therapy available; they have many uses including both curative and symptom-reducing functions [10]. Anti-cancer therapy has advanced in the past years with the developments of targeted therapies and immunotherapies, which can be used as monotherapy or adjunctively with chemotherapy [11]. Toxic erythema of chemotherapy, linear IgA bullous dermatosis, hand–foot skin reaction, bullous lichenoid drug eruption, and Stevens–Johnson syndrome (SJS)/toxic epidermal necrolysis (TEN) and SJS-like eruptions are bullous DAEs that have been associated with chemotherapy.

Targeted therapies, such as kinase inhibitors, monoclonal antibodies, and antibody-drug conjugates, aim to inhibit molecular pathways involved in tumor growth and maintenance [12,13]. Targeted therapies are typically used in tumors with known pathogenesis or survival mechanisms, for example, tumors with targetable driver mutations or specific proteins known to be involved in tumorigenesis [14]. One benefit of targeted therapies is the specific nature of their effects, which often serves to minimize adverse events as compared to cytotoxic chemotherapy [15]. Targeted therapies have been shown to induce rapid tumor regression. However, resistance can be induced by pathway bypass or mutations in target molecules at high rates. For example, up to 46% of patients receiving epidermal growth factor receptor inhibitors have developed resistance; favorable responses may be short lived [16]. Hand–foot skin reaction, toxic erythema of chemotherapy, SJS/TEN and SJS/TEN-like eruptions, and non-specific bullous drug eruption are bullous DAEs that have been associated with targeted therapy [17,18]. 

Immunotherapy aims to stimulate a host immune response to cause tumor destruction. Types of immunotherapy that will be discussed include immune-checkpoint inhibitors (ICI) targeting programmed cell death 1 (anti-PD1), programmed cell death ligand 1 (anti-PD-L1), cytotoxic T lymphocyte antigen 4 (anti-CTLA-4), and other ligand (anti-CD274 and anti-CD137) axes, as well as toll-like receptor (TLR) 8 agonists [19,20]. Tumor cells may become resistant to innate cytotoxic T cell induced-apoptosis; ICI serve to disinhibit T cells to restore host immune ability to destroy tumor cells [21]. TLR agonists, such as TLR7 and TLR8, activate transcription factors to induce cytokine production to subsequently induce a response against cancer cells [22]. Immunotherapy harnesses the host immune system and has the potential to treat a broad range of cancers with a durable effect on outcomes [20]. Most types of bullous DAEs that will be discussed in this review, with the exception of toxic erythema of chemotherapy and hand–foot skin reaction, have been associated with immunotherapy.

We will divide cancer therapy DAEs into cutaneous and systemic drug eruptions, summarized in Table 1. For the purpose of this review, cutaneous DAE are defined as those involving only the skin with no existing or potential mucous membrane involvement or systemic associations. Systemic DAE are defined as those with both skin and existing or potential mucous membrane involvement or systemic involvement of various organs, such as blood, kidney, liver, etc. In addition, patients with systemic drug eruptions may have signs of cough, malaise, fever, and myalgias. When considering bullous eruptions, in particular, it is important to distinguish between cutaneous and systemic DAE due to the increased morbidity and mortality associated with some systemic bullous eruptions, namely Stevens–Johnson syndrome/toxic epidermal necrolysis (SJS/TEN) [23]. The differentiation between cutaneous and systemic eruptions is important as it provides a framework for practical medical decision-making.

As defined above, cutaneous bullous drug eruptions include vesiculobullous eczema, hand–foot skin reaction, and toxic erythema of chemotherapy. Systemic bullous eruptions include bullous pemphigoid, bullous lichenoid drug eruption, lichen planus pemphigoides, pemphigus vulgaris, bullous erythema multiforme, linear IgA bullous dermatosis, bullous lupus erythematosus, Stevens–Johnson Syndrome (SJS)/toxic epidermal necrolysis and SJS-like eruptions, and non-specific bullous drug eruption.

In patients with cutaneous drug eruptions from anti-cancer treatment the Common Terminology Criteria for Adverse Events (CTCAE) grading system for all types of drug-induced DAE, cutaneous and systemic, is used as defined in Table 2. Severity of DAEs can be subtyped as low grade (CTCAE grades 1–2) and high grade (CTCAE grade ≥3). Treatment of DAEs can be based on CTCAE grading (Figure 1). The proposed pathogenesis of cutaneous and systemic bullous DAE are summarized in Table 3. 

In this review, we will review the pathogenesis, diagnosis, grading, and an evidence-based approach to management of bullous DAEs associated with anti-cancer therapies. 

## 2. Literature Search Methods

A literature review of bullous DAE induced by anti-cancer therapy was conducted using PubMed/MEDLINE. Search terms included “bullous,” “cutaneous adverse events,” “blister,” “dermatologic adverse events,” “Steven Johnson syndrome/toxic epidermal necrolysis,” “anti-cancer therapy,” “chemotherapy,” “immunotherapy,” and “targeted therapy”. Bullous DAE included were selected based on clinical presentation and relevance as determined by leading experts in the field. Case reports, reviews, and original research articles were included between 1990 and 2022, with greater than 50% of studies cited published within the last 5 years. 

## 3. Cutaneous Bullous Dermatologic Adverse Events

### 3.1. Vesiculobullous Eczema 

Anti-cancer therapy-induced eczema is not uncommon, with an estimated incidence of 17% following ICI therapy, specifically nivolumab. Acute, severe forms of eczema have manifested with bullous features in a few reports [8,33,48,49,50]. Vesiculobullous eczema is graded as CTCAE eczema grades 1–3.

There has been one case of vesiculobullous eczema reported 3 months after nivolumab initiation. The patient clinically presented with prodromal erythematous plaques on the dorsum of the hands followed by a diffuse, scaly, bullous eruption involving the upper and lower extremities [8]. Pathogenesis related to anti-cancer-therapy-induced vesiculobullous eczema has not yet been postulated. However, idiopathic bullous eczema is hypothesized to result from over-expression of aquaporin 3 and aquaporin 10 in keratinocytes throughout the mid and upper epidermis, resulting in epidermal fissuring and subsequent vesicle formation secondary to cutaneous water and glycerol outflow [51]. 

Histopathology may reveal spongiotic dermatitis along with lymphocytic dermal infiltrates; Civatte bodies and parakeratosis may also be present. DIF can be utilized to exclude bullous pemphigoid (BP) [8].

Vesiculobullous eczema may be treated with topical and oral corticosteroids; PD-1 inhibitor therapy may be held for high grade eruptions. Vesiculobullous eczema has reportedly resolved following nivolumab cessation; however, mild eczema has been noted to persist for months [8]. 

### 3.2. Hand–Foot Skin Reaction (HFSR), Bullous Type 

Hand–foot skin reaction (HFSR) is a painful eruption of sharply demarcated hyperkeratotic erythematous papules and plaques on pressure points of palmar-plantar surfaces and distal phalanges; when moderate to severe (grade ≥ 2), the manifestations are often bullous. Blisters are often tender and heal into hyperkeratotic inflamed calluses [7,52]. HFSR has been reported in about 30% of patients on targeted therapies, most commonly vascular endothelial growth factor (VEGF) inhibitors, including kinase inhibitors such as sorafenib, sunitinib, axitinib, pazopanib, and regorafenib, or anti-angiogenetic drugs, such as vemurafenib and dabrafenib [52,53,54,55,56,57,58].

Anti-cancer induced HFSR incidence and severity is typically dose-dependent [7,52]. HFSR usually causes a localized erythematous reaction [7]. HFSR is typically graded as CTCAE bullous dermatitis grade 2–3.

Pathogenesis of HFSR is still not known. Various theories have been proposed. HFSR is postulated to be the result of direct blockade of VEGFR, PDGFR, and EGFR in healthy tissue. [26,27,28,29]. PDGFR, in particular, is highly expressed in the eccrine gland ductal epithelium. Eccrine excretion of the inciting drug is postulated to cause direct dermal toxicity and/or inhibit receptors, such as PDGFR, leading to impaired wound healing especially in frictional areas [29]. High friction and pressure areas, such as on the palmar-plantar surfaces as well as on the elbows and knees, are constantly exposed to recurrent microtrauma; thus, at these locations the lesions are often higher grade due to their limited vascular supply [7,29]. Further, some authors have suggested that HFSR may be equivalent to a Koebner phenomenon, which is the development of new skin lesions secondary to trauma [30].

Although this is typically a clinical diagnosis and does not require skin biopsy, bullous HFSR histopathology usually demonstrates characteristic keratinocyte damage in the form of vacuolar degeneration, keratinocyte apoptosis or necrosis, and intracytoplasmic eosinophilic bodies; these features often cause intraepidermal cleavage. There may be intraepidermal blisters in the stratum malpighii. Additional features that may be present include dyskeratotic cells, papillomatosis, epidermal acanthosis, or parakeratosis [7].

HFSR treatment ideally begins with prophylactic techniques prior to initiating anti-cancer therapy, such as a hand and foot skin exam to identify predisposing hyperkeratotic skin. Patients with hyperkeratotic skin on anti-cancer therapies implicated in the development of HFSR may benefit from wearing thick gloves and socks to prevent friction or trauma to palmar-plantar surfaces [52].

If bullous HFSR develops, treatments should be based on grade of severity. Treatment starts with emollients and lifestyle changes to reduce palmar-plantar friction and can escalate to topical corticosteroids, topical keratolytic agents, and if needed, systemic pain medications. Lastly, patients may benefit from anti-cancer dose modifications or discontinuation [52].

### 3.3. Toxic Erythema of Chemotherapy (TEC)

Toxic erythema of chemotherapy (TEC) is a common diagnosis encompassing a spectrum of cutaneous eruptions secondary to the use of anti-cancer therapy, including palmar-plantar erythrodysesthesia (PPE) and severe bullous flexural dermatitis (SBFD) [59,60,61]. TEC is also known as malignant intertrigo when involving the intertriginous skin. Diagnosis is of exclusion and based on clinical presentation, histologic findings, and known associations [5]. TEC is graded as CTCAE palmar-plantar erythrodysesthesia syndrome grades 1–3.

A number of anti-cancer drugs have been reported in association with TEC. The most commonly reported drugs include cytotoxic chemotherapies with an overall incidence of 3–64%, most commonly doxorubicin and cyclophosphamide; others include, paclitaxel, gemcitabine, decitabine, cytarabine, daunorubicin, methotrexate, cyclosporine, FOLFIRI (leucovorin calcium, 5-fluorouracil, and irinotecan), and vinorelbine [31,49,62,63]. Antibody drug conjugates such as brentuximab vedotin and enfortumab vedotin have also been reported to cause TEC [5,32,33,59,64,65]. It is important to note that bullous TEC secondary to enfortumab vedotin can present with widespread blistering and appear similar to TEN [64]. Few cases have been reported with no to minimal mucosal involvement; as such, it is included as a cutaneous DAE [64,66,67,68]. DAE onset typically ranges from days to months after anti-cancer therapy initiation [31,33,69].

TEC typically presents as red-purple patches and plaques, with bullae and erosions in severe cases, favoring the hands, feet, and intertriginous skin [59,64]. TEC typically spares the mucous membranes and lacks confluent erythroderma, which helps to differentiate it from SJS/TEN in otherwise ambiguous cases [59]. TEC may initially present with tingling and burning paresthesia as well as erythema in palms, fingers, and soles. Symptoms classically progress to involve edema, blisters, and ulcerations [62].

Pathogenesis of cytotoxic chemotherapy therapy-induced TEC is likely related to drug accumulation in eccrine sweat glands and subsequent local toxicity [31]. The pathogenesis of TEC secondary to enfortumab vedotin therapy is postulated to be the same mechanism as cytotoxic chemotherapy but more specifically inducing toxicity by depositing the cytotoxic monomethyl auristatin E (MMAE) in tissues expressing nectin-4, such as the skin. Enfortumab vedotin induces apoptosis of keratinocytes expressing nectin-4, causing dysfunctional cell-cell adherence and bullae formation [32,33]. 

Laboratory studies in TEC are typically within normal limits, and apparent lab abnormalities are typically attributed to the chemotherapy itself [33]. Though diagnosis can be made clinically and biopsy is rarely indicated, on histopathology, TEC presents with thickened epidermis with dyskeratosis and suprabasalar acantholysis as well as eccrine duct atypia. Interface dermatitis with necrotic keratinocytes and focal eccrine gland/duct necrosis may also be seen [5]. Increased mitotic figures without evidence of epidermal regeneration, squamatization of the basal layer, and syringosquamous metaplasia and the presence of only scattered necrotic keratinocytes suggest TEC over TEN histologically [59]. Histopathologic features of TEC include parakeratosis, epidermal acanthosis, papillomatosis, and vacuolar degeneration. The granular layer may be absent. Vasodilation and perivascular mononuclear cell infiltrates may also be present in the dermis [7]. Histopathology of bullous TEC lesions induced by enfortumab vedotin, specifically, may uniquely reveal disrupted cytoskeletons, as evidenced by abnormal and arrested mitoses as well as apoptotic cells with minimal dermal lymphocytic infiltration and epidermal dysmaturation [48]. DIF may demonstrate IgG and C3 cell surface deposits in the epidermis corresponding with the location of nectin-4, as well as intermittent linear deposition of IgM at the dermo-epidermal junction (DEJ) [32]. 

TEC does not typically require anti-cancer therapy discontinuation [70]. Rather, it is a toxicity and requires symptomatic treatment including treatment with topical corticosteroids, topical lidocaine, cold compresses, or oral corticosteroids; IVIg can be used in severe cases [5,31,33,59]. Some patients may require oral pain medications [31]. Dexamethasone administered in conjunction with cytotoxic chemotherapy has been shown to reduce the risk of developing TEC [71].

## 4. Systemic Bullous Dermatologic Adverse Events

### 4.1. Bullous Pemphigoid (BP)

Bullous pemphigoid (BP) is classically caused by autoantibodies to BP180 and BP230, two basement membrane hemidesmosome proteins, leading to the development of localized or generalized tense bullae, most commonly in the elderly [72]. The development of BP has been associated with primary cancers, including melanoma and non-small cell lung cancer most notably; however, BP has also been associated with ICI targeting PD1/PDL1 [34,73,74]. The incidence of BP in patients taking ICI (ICI-BP) is about 1%; BP is well established as the most common bullous eruption secondary to ICI (Figure 2) [3,34,35]. ICI-BP onset is classically delayed, occurring usually 4 months after ICI therapy initiation, with some cases developing after 1.5 years of therapy or even after ICI discontinuation [4,74,75]. Prodromal symptoms of ICI-BP may include generalized pruritus, followed by the formation of macular or urticarial lesions, and then followed by the development of tense bullae on the extremities and torso that are filled with either serous or hemorrhagic fluid. Oral mucosal involvement is reportedly present in up to 40% of ICI-BP, contrary to primary BP which has mucosal involvement in only 19% of patients [3,34,73,76,77,78]. BP is graded as bullous dermatitis grade 1–5 [4].

There are multiple theories regarding the pathogenesis of ICI-BP. The most well-established theories include activation of antibody-secreting B cells, inhibition of immunosuppressive regulatory T-lymphocytes, cross-reaction between anti-BP180 antibodies since BP180 is expressed by many tumor cells, or the triggering of clinically undetectable emerging BP by ICI [3,8,34,35,36,37]. Although patients with HLA-DQB 1*03:01, a major histocompatibility complex class-II allele, have been shown to be more likely to develop primary BP, it is unclear if such genetic predispositions hold true for ICI-BP [74,79]. It is also unknown if people who develop ICI-BP have BP180 antibodies prior to ICI initiation [35]. Future research assessing what types of patients may be predisposed to developing ICI-BP could be helpful for screening and monitoring purposes. ICI-BP is a positive predictor for ICI cancer response, likely due to a robust host immune response [80]. 

Histopathology, immunofluorescence, and ELISA of ICI-BP are all similar to classic autoimmune BP [3]. Histopathology usually demonstrates subepidermal clefting with eosinophils and fibrin with lymphocytes, eosinophils, and scattered neutrophils composing a band-like dermal infiltrate. DIF demonstrates linear deposition of IgG and C3 at the DEJ. ELISA demonstrates BP180 positivity, and in some cases, BP230 as well [1,3,34,35,76,77].

Treatment is based on the grade of ICI-BP. The treatment approach recommended is to continue ICI treatment in grade 1 DAEs and provide BP-specific medications; however, if grade 2 or higher, anti-cancer therapy should be held until the DAE resolves to grade 0 or 1 [81,82]. Medications include topical corticosteroids, oral corticosteroids, and systemic steroid-sparing drugs such as methotrexate, dapsone, azathioprine, mycophenolate mofetil, omalizumab, dupilumab, rituximab, or IVIg [4,83,84,85,86]. If BP persists, then the ICI can be held or discontinued; however, ICI cessation alone has not been proven to be curative [4,35,83,87,88,89,90]. ICI-BP is often severe and challenging to treat. One review reported that 76% of patients who developed ICI-BP required ICI discontinuation; however, 19% of these patients continued to have BP recurrences 3–12 months following discontinuation [74]. 

Given the importance of anti-cancer therapy, it is beneficial to identify ICI-BP early and to manage symptoms to reduce treatment disruption. Since generalized pruritus may be the only prodromal feature of ICI-BP, providers may consider ordering DIF in patients experiencing new onset pruritus following ICI initiation. This will serve to aid in early diagnosis and prophylactic treatment prior to diffuse bullous eruption to minimize anti-cancer treatment interruptions [74].

### 4.2. Bullous Lichenoid Drug Eruptions (BLDE)

Bullous lichenoid drug eruptions (BLDE) include bullous lichen planus (BLP) and bullous generalized lichenoid eruptions, severe forms of lichen planus (LP) that can be familial or drug-induced. These are rarer forms of DAE compared to BP that can occur secondary to ICI and chemotherapy. BLDE presents with an initial lichenoid, maculopapular rash, often with keratotic, purple papules and plaques and with diffuse pruritus followed by the development of tense vesicles and bullae. BLDE typically presents as ill-defined erosions and vesicles on the legs and trunk, sparing the oral and genital mucosa and with usually a negative Nikolsky sign [91]. In severe cases, the Nikolsky sign may be positive [39]. While the oral mucosa is typically spared of bullae and erosions in BLDE, there may be mucosal involvement and Wickham striae in the BLP subtype [4,39,90,91,92]. Chemotherapy drugs and ICI have been implicated in bullous lichenoid eruptions, including nivolumab, ipilimumab, and pembrolizumab [8,35,39,90,91]. The latency of BLP after ICI (ICI-BLP) initiation varies, with reports ranging from 3 weeks to 12 months [4,93]. For bullous generalized lichenoid eruptions a longer lag period has been reported (3 to 20 months) [39]. The epidemiology is difficult to characterize due to the rarity of eruptions. BLDE is typically graded as CTCAE bullous dermatitis grade 1–5.

CD4+ and CD8+ T cells are thought to be primary drivers in the pathogenesis in BLDE [91]. The pathogenesis of BLDE has been theorized to be due to bullae developing at the site of exuberant lichenoid dermatitis [38]. In addition, some have suggested that the pathogenesis of ICI-induced BLDE is similar to that of SJS/TEN, involving apoptosis of basal keratinocytes secondary to activation of CD8+ T cells by the perforin/granzyme pathway [39]. 

Histopathology of BLDE is significant for lichenoid interface dermatitis, focal hypergranulosis, eosinophils, and focal necrotic keratinocytes [39,91]. Reports have also shown focal subepidermal clefting, prominent lymphocytic infiltrate, sawtooth acanthosis, basal vacuolar degeneration, hyperkeratosis, orthokeratosis, and parakeratosis, all of which differentiate BLDE from other bullous eruptions [39]. DIF will typically be negative or may show focal IgM deposition and C3 colloid bodies at the DEJ in a non-linear pattern. ELISA is typically negative for both BP180 and BP230 [4,8,39,63,91,92,93]. Laboratory workup may show nonspecific elevations of ESR, CRP, and procalcitonin, as well as hypoalbuminemia [39].

It is recommended to continue ICI treatment and provide BLDE-specific medications for grade 1 disease [93]. Grade 1BLDE are treated similarly to LP; first-line treatment includes topical and/or oral corticosteroids. Grade 2–3 and above requires more aggressive therapy. There are no clear guidelines for steroid-sparing therapies; however, dupilumab, cyclosporine, infliximab, rituximab, IVIg, systemic acitretin, and PUVA therapy have been used with varying efficacy [4,91,93]. If BLDE is persistent, ICI can be held or discontinued. However, one review of six cases reported that 83% of ICI-BLP, specifically, were responsive to treatment with one or multiple of the aforementioned therapies [93]. 

### 4.3. Lichen Planus Pemphigoides (LPP)

Lichen Planus Pemphigoides (LPP) is considered distinct from BP and BLP, although clinically the three share similar characteristics, such as bullous and lichenoid features. In both BLP and LPP, bullae occur on lichenoid plaques, but in LPP bullae may also develop on previously unaffected skin and oral mucosal involvement in about half the cases [9]. Anti-cancer-therapy-induced LPP is rare but has been reported in patients on ICI such as pembrolizumab, nivolumab, atezolizumab, and tislelizumab [9]. LPP has developed on average 6 months after ICI initiation; however, onset ranges from 4 weeks to over 1.5 years [9]. LPP is typically graded as CTCAE bullous dermatitis grade 2–3.

The pathogenesis of LPP is thought to be due to epitope spreading within a lichenoid rash. ICI may cause lichenoid dermatitis; this interface dermatitis subsequently leads to BP180 exposure at the DEJ, allowing the host immune system to develop antibodies targeting these exposed BP180 self-antigens. This develops into epitope spreading, leading to autoimmune bullous progression of lichenoid lesions [40,41,42].

Histopathology of LPP lesions demonstrate features of BLP and LP. Often there is subepidermal clefting with lymphocyte-rich infiltrate [36]. Features such as colloid bodies and focal vacuolar degeneration, as in LP, have been noted as well [4,9]. Bullae in LPP that arise from pre-existing lichenoid plaques differ from BLP because LPP bullae often demonstrate additional eosinophilic and neutrophilic infiltrate [9]. DIF may show IgG and C3 at the DEJ [4,9,36]. Indirect immunofluorescence (IIF) has positively identified epidermal basement membrane zone proteins [9]. Although LPP may be BP180 positive, LPP BP180 has a distinct NC16A domain C-terminal region 4 [9,36]. ELISA is usually positive for BP180 [36].

LPP is often challenging to treat particularly in cases with greater severity. Patients with CTCAE grade 2 or higher LPP should have their ICI held until the LPP severity decreases to grade ≤ 1 [37]. Treatment should target both the bullous pemphigoid component and lichenoid components of the eruption. First-line therapy includes topical and/or systemic corticosteroids, often in combination with agents traditionally used for BP, such as rituximab and IVIg. Acitretin, sirolimus, and dapsone have also have varying efficacy [36]. 

### 4.4. Pemphigus Vulgaris (PV)

Pemphigus vulgaris (PV) is a bullous autoimmune disease which presents as painful, nonpruritic flaccid bullae that can affect the skin and mucous membranes due to loss of cell adhesion in the epidermis. It has rarely been associated with anti-cancer drug use; thus, the incidence is difficult to measure. It is important to distinguish PV from paraneoplastic pemphigus (PNP). Nivolumab, an ICI, is the only anti-cancer therapy reported in the literature to cause a variant of pemphigus to date. Two cases of PV were reported following nivolumab initiation [43,94]. PV is graded as CTCAE bullous dermatitis grade 1–5.

One reported case of typical PV was found to be a recurrence in a patient with a history of PV diagnosed 15 years prior, whose disease was in remission for 7 years [43]. A case of atypical PV was reported in a patient 2.5 weeks after discontinuation of nivolumab therapy in a patient with no prior history of autoimmune disease [94]. One of two reported cases involved the oral cavity, and both cases lacked fever or other prodromal features [43,94]. 

The pathogenesis of PV is theorized to be an immune-mediated T-cell reaction secondary to nivolumab, triggering onset or recurrence in susceptible patients [43]. Spontaneous PV is thought to be the result of circulating IgG to desmoglein-3 and sometimes desmoglein-1, causing dissociation at the epidermal desmosomes and subsequent acantholysis. It is possible that nivolumab causes an upregulation of these antibodies through a generalized increase in immune function, triggering PV [43].

Histopathologic findings may demonstrate suprabasal clefting with fibrin, acantholytic cells, eosinophils, and neutrophils in the lumen of vesicles. Eosinophils can also be found in the upper dermis. On DIF, intercellular deposits of IgG on keratinocyte surfaces may be seen in the epidermis [43,94]. Autoimmune PV is typically positive for desmoglein-1 and -3. PNP is typically positive for envoplakin and periplakin. In patients with ICI-PV, on ELISA, desmoglein-3 has reportedly been positive with negative envoplakin and periplakin, ruling out PNP [43,94]. ICI-PV may be desmoglein-1 and 3 positive, anti-desmocollin-2 and 3 antibodies positive, and negative for BP180 and BP230 [43,94].

In both reported cases, PV was successfully treated without the discontinuation of nivolumab. Treatment included topical corticosteroids, oral corticosteroids, intravenous immunoglobulin, sirolimus, mycophenolate mofetil, and/or oral methotrexate with resolution of symptoms [43,94]. While these studies did not utilize rituximab, this is also an option for treatment of PV. 

### 4.5. Bullous Erythema Multiforme (BEM)

Bullous erythema multiforme (BEM) has been reported to be induced by ICI, such as PD-1 inhibitors nivolumab and pembrolizumab. BEM clinically presents with diffuse, flaccid bullae, painful erythematous plaques, and targetoid lesions, some with central necrosis, widespread on the body [95,96,97,98]. Oral mucosal ulcerations may also be present [96,98]. 

The incidence of ICI-BEM is estimated to be 3–4%, with most being reported as single cases in review articles or case reports. BEM clinically presents between 3 weeks and 38 months after ICI initiation; however, most occur within 3 months [90,95,96,97]. BEM is graded as CTCAE erythema multiforme grades 1–5.

The pathogenesis of BEM is theorized to be a severe immune reaction to antigens as a result of CD4+ and CD8+ T-cell imbalance in the host. Such imbalance likely has multiple contributing mechanisms. One suspected pathway involves increased expression of Fas ligand on T cells in response to nivolumab, causing increased keratinocyte apoptosis. Another theory involves increased differentiation of immature T cells expressing CTLA-4 in response to ipilimumab, causing a hypersensitivity loop of activated T cells to a triggering antigen. Similar to non-ICI induced BEM, autoreactive T cells and associated cytokines may lead to the pathologic findings of the disease state [44].

BEM is usually histologically characterized by blisters at the DEJ with vacuolar degeneration accompanied by eosinophilic and predominantly T-cell lymphocytic infiltrate [44,95,96,97]. Dyskeratotic keratinocytes may also be present [96]. 

Treatment is primarily supportive. Topical steroids may be used for symptom management. Systemic corticosteroids, dapsone, azathioprine, or thalidomide are typically used as next-line therapies. Cyclosporine, IVIg, and infliximab have also been used [44,90,96,98]. For more high-grade cases, anti-cancer therapy discontinuation may be indicated, and in many cases, therapy is not rechallenged [95,98]. 

### 4.6. Linear IgA Bullous Dermatosis (LABD)

Linear IgA bullous dermatosis (LABD) is a rare autoimmune disorder characterized by subepidermal blistering and IgA deposition in a linear pattern along the basement membrane [99]. Two classes of anti-cancer drugs have been implicated in the development of linear IgA bullous dermatosis: antimetabolite chemotherapy (i.e., gemcitabine), and ICI targeting PDL-1 (i.e., durvalumab and atezolizumab). There have been three cases reported, induced by gemcitabine, durvalumab, and atezolizumab [35,100,101]. The durvalumab case, however, is complicated by the initiation of vancomycin, a commonly implicated drug in LABD, one week prior to eruption [35]. The estimated incidence of LABD ranges from 0.2 to 2.3 per 1 million per year, but this includes both idiopathic and acquired cases across all ages [45]. 

Drug-induced LABD presents 1 day to 2 weeks after exposure to the offending agent and is a clinically heterogeneous disease [100]. It can present similar to dermatitis herpetiformis with symmetric, bullous, herpetiform lesions on the trunk and upper extremities; however, it has also been reported as an urticarial eruption mimicking EM, BP, and even SJS/TEN [100]. LABD may present with oral mucosal involvement [102]. The bullae can coalesce into annular plaques with a targetoid appearance in some cases [101]. LABD is graded as CTCAE bullous dermatitis grade 1–5.

The pathogenesis of anti-cancer-drug-induced LABD has not been postulated in previous studies. However, spontaneous LABD has been well-characterized and involves circulating IgA anti-basement membrane zone antibodies directed against the 97 kDa portion of BP180 in the lamina lucida. There have also been reports of LABD occurring in patients with a number of malignancies, including lymphoproliferative disorders and thyroid, bladder, colon, renal, and esophageal cancers [45]. Thus, it is possible that the cases of LABD reported were predisposed by their underlying malignancy with clinical features triggered by inciting drugs, or secondary to the drugs themselves. As such, this is an avenue for further study. 

Histopathology of LABD usually includes subepidermal blisters with papillary abscesses containing predominantly neutrophils and occasionally eosinophils [100]. However, this is not always the case; similar to the heterogeneous presentation, histopathology can also mimic other conditions, so immunopathologic studies are crucial for diagnosis. DIF will reveal IgA in a linear pattern with or without IgG and C3 at the DEJ. ELISA may be positive for IgA antibodies to BP180 [100]. 

LABD is treated with topical corticosteroids and systemic therapy, such as oral corticosteroids or dapsone based on severity [100]. Anti-cancer therapy cessation may be required. In general, symptoms tend to remit within 2–6 weeks after discontinuation of the offending agent [45]. 

### 4.7. Bullous Lupus Erythematosus (BLE)

Bullous lupus erythematosus (BLE) is a rare autoimmune disease characterized by subepidermal blisters secondary to autoantibodies against type VII collagen. BLE presents as a widespread, often photo-distributed eruption of tense vesicles and bullae that can affect both the skin and mucous membranes [6]. The incidence of BLE is about 0.2 cases per million per year, encompassing both idiopathic and drug-induced presentations [103]. BLE is graded as CTCAE bullous dermatitis grade 1–5.

ICI, specifically, nivolumab, has been associated with an exacerbation of BLE in a patient with possible paraneoplastic lupus from lung adenocarcinoma. This patient presented with a nonspecific rash shortly after diagnosis of his cancer, which was followed by intermittent flares until cycle 8 of nivolumab, at which time he developed a bullous lupus eruption on his extremities, oral mucosa, and genitals [6]. 

The pathogenesis of drug-induced cutaneous BLE is speculative, and available data suggests that different drugs likely have different underlying mechanisms. Thus, it is difficult to characterize the relation of anti-cancer therapy to BLE. 

Histopathology of BLE can show subepidermal blisters and neutrophilic infiltrate at the DEJ with subepidermal clefting, numerous apoptotic keratinocytes, and basal keratinocyte vacuolization. Increased mucin and micro-abscesses may be seen in the papillary dermis [103]. DIF will reveal IgG and C3 in a linear pattern with IgM and IgA in a granular pattern along the DEJ [103]. The absence of eosinophilia on biopsy helps to differentiate BLE from dermatitis herpetiformis, LABD, and epidermolysis bullosa [103]. Laboratory evaluation may show a positive ANA with a speckled pattern, with one study showing positive anti-Ro/SSA, anti-RoSSA52, and p-ANCA, as well as an elevated ESR [6].

Treatment of drug induced BLE may involve discontinuation of the causative drug and oral corticosteroids [6]. Dapsone is considered first-line therapy for BLE and can likely be used for drug-induced BLE, as well. 

### 4.8. Stevens–Johnson Syndrome (SJS)/Toxic Epidermal Necrolysis (TEN) and SJS-Like Eruptions

SJS/TEN is a rare, potentially life-threatening cutaneous blistering disorder that can be a complication of many anti-cancer therapies, including chemotherapy, immunotherapy, and targeted therapy. The epidemiology of SJS/TEN associated with anti-cancer therapy has not been well-defined in the literature. In general, severe bullous DAE, including SJS/TEN, bullous lichenoid drug eruptions, and drug-induced BP account for <6% of all DAE secondary to anti-cancer therapy [39]. SJS/TEN-like eruptions mimic SJS/TEN but vary in clinical course, severity, and treatment response; these reactions typically present and evolve with weeks to months of exposure to causative drug rather than acutely and resolve slowly over weeks with a more benign course than true SJS/TEN (Figure 3) [4]. 

A number of anti-cancer drugs have been linked with SJS/TEN. Causative chemotherapy drugs include methotrexate, alkylating agents, thalidomide, docetaxel, mithramycin, doxorubicin, L-asparaginase, cytarabine, and gemcitabine with concurrent radiation [104,105,106,107,108,109,110,111]. Targeted therapies associated with SJS/TEN include BRAF inhibitors and drugs from the receptor tyrosine kinase inhibitor family, namely EGFR, BCR-Abl, and KIT inhibitors, as well as combination therapy and monoclonal antibodies such as rituximab [46]. Immunotherapies include ICI such as PD-1, PD-L1, and CTCLA-4 inhibitors [8,39,46]. SJS/TEN-like eruptions are more likely to occur following PD-1/PD-L1 therapy than true SJS/TEN [4].

SJS/TEN usually initially presents with a flu-like prodrome that progresses to a painful maculopapular rash with blistering eruptions on dusky purpuric macules or atypical targetoid patches and with erosions of multiple mucous membranes [8,104]. This can be life-threatening in instances of mucosal involvement causing tracheobronchial detachment, digestive system involvement, severe ocular involvement, and significant skin detachment [112]. Skin examination will typically reveal a positive Nikolsky sign and is followed by eventual desquamation resembling a second-degree burn [8,104,113]. These diseases are differentiated by body surface area (BSA) involvement, with SJS involving <10% BSA, SJS/TEN involving 10–30% BSA, and TEN defined as >30% BSA. Laboratory evaluation during workup of SJS/TEN will typically reveal elevations in nonspecific inflammatory markers. There is no single laboratory abnormality that is pathognomonic for SJS/TEN. [104,113]. Compared to SJS/TEN, which presents acutely, SJS/TEN-like eruptions often develop over the course of weeks to months or late in the course of treatment ranging up to 420 days following PD-1/PD-L1 therapy initiation. SJS/TEN-like reactions are also less likely to have significant systemic involvement; fever and ocular involvement is much rarer, estimated to occur in 8% of patients [4]. SJS/TEN and SJS/TEN-like eruptions are graded individually per CTCAE grading criteria as either SJS grade 3–4 or TEN grade 4. 

SJS/TEN is a delayed-type hypersensitivity reaction in which cytotoxic T cells generate and release granulysin, which leads to disseminated keratinocyte death. It is thought that with ICI, specifically, PD-1, PD-L1, or CTCLA-4, inhibition leads to impaired T cell homeostasis in the skin and loss of protection from skin autoimmunity, leading to cytotoxic inflammatory reactions [8,39]. However, with EGFR inhibitors, it is theorized that irreversible inhibition of EGFR leads to interference of epidermal differentiation and re-epithelialization which leads to extensive erosions and the clinical appearance of SJS/TEN [46]. 

Histopathology of SJS/TEN and SJS/TEN-like reactions typically reveals full-thickness necrosis of keratinocytes with subepidermal clefting often with sparse mononuclear dermal infiltrate and CD8+ T cells in the epidermis and at the DEJ [4,34]. ELISA and DIF are not needed for diagnosis and are usually negative. However, if there is concern for PNP, which presents similarly and is classically associated with rituximab, DIF can be used to exclude PNP.

SJS/TEN requires immediate, early intervention given its high mortality rate of 10–50% [46,114]. Typically, the suspected inciting drug should be immediately discontinued. Further management options include etanercept, cyclosporine, and IVIg. Systemic steroids have been associated with increased mortality and are typically avoided [8,46,104,115,116,117]. Supportive wound care is a mainstay of therapy. Of note, SJS/TEN-like reactions are often milder and have a more favorable treatment response as compared to true SJS/TEN [4]. 

### 4.9. Non-Specific Bullous Drug Eruption (NSBDE)

Non-specific bullous drug eruption (NSBDE) is an umbrella term that encompasses otherwise unspecified bullous eruptions in response to a drug based on clinical features such as timeline and resolution following drug withdrawal. As this is a diagnosis primarily of exclusion, it is difficult to estimate the incidence of such reactions, particularly to anti-cancer therapy [118]. While these do not have a pathognomonic clinical or histologic presentation, it is important to be aware that not all bullous drug eruptions will fit into the aforementioned categories, and further exploration of these eruptions is critical for furthering our understanding of other bullous DAEs. 

## 5. Limitations

The main limitation of this review is that many anti-cancer-therapy-induced bullous DAE are published in case reports or case series. There are no double-blinded randomized-control trials assessing bullous DAE following anti-cancer therapy initiation. Furthermore, some bullous DAEs have only been reported in a few patients and thus it is not certain whether the patient was going to develop a bullous disease regardless of anti-cancer therapy. Many patients on anti-cancer therapy are on concurrent medications; it cannot be proven with certainty that all bullous DAE associated with anti-cancer therapy were causative cases. Pathogenic mechanisms discussed in this review are based on clinical opinions proposed in the literature. Lastly, this review is not a systematic review which limits the degree of objectivity. 

## 6. Conclusions

Anti-cancer therapy improves outcomes for cancer patients; however, many classes of anti-cancer therapy have been implicated in the induction of bullous DAE, leading to reduced patient quality of life and in some cases discontinuation of life-prolonging or palliative therapy. ICI have been found to be the most commonly implicated in the development of bullous DAE, likely due to their immune-enhancing effects. Cytotoxic chemotherapies have been reported less frequently, with their direct cytotoxic effects on the DEJ causing bullae formation, followed by few reports of targeted therapy-induced bullous DAE with less understood mechanisms.

Bullous DAE may be limited to the skin or have systemic involvement with greater risk of morbidity and mortality. We present the epidemiology, diagnosis, pathogenesis, and management of bullous DAE secondary to anti-cancer therapies to enable clinicians to more optimally manage these patients. 

## Figures and Tables

**Figure 1 biomedicines-11-00323-f001:**
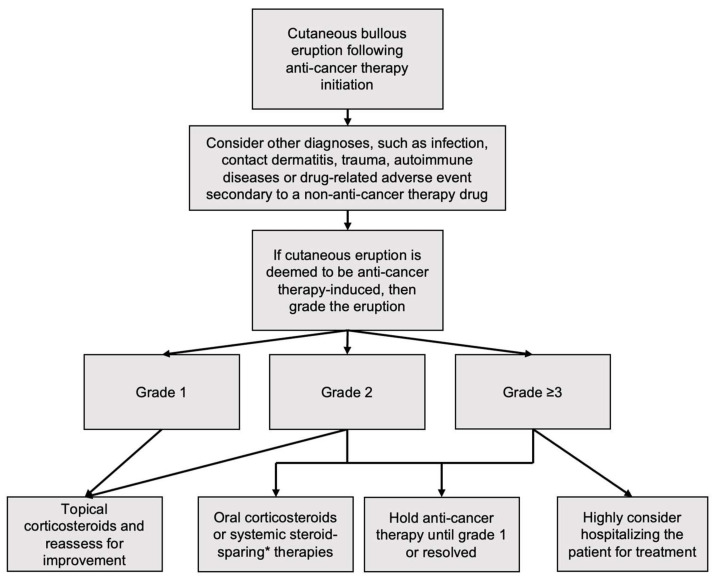
Proposed treatment algorithm following anti-cancer-therapy-induced bullous dermatologic adverse events [24,25]. * Such as etanercept, tocilizumab, methotrexate, dapsone, azathioprine, mycophenolate mofetil, omalizumab, dupilumab, rituximab, or IVIg depending on reaction type.

**Figure 2 biomedicines-11-00323-f002:**
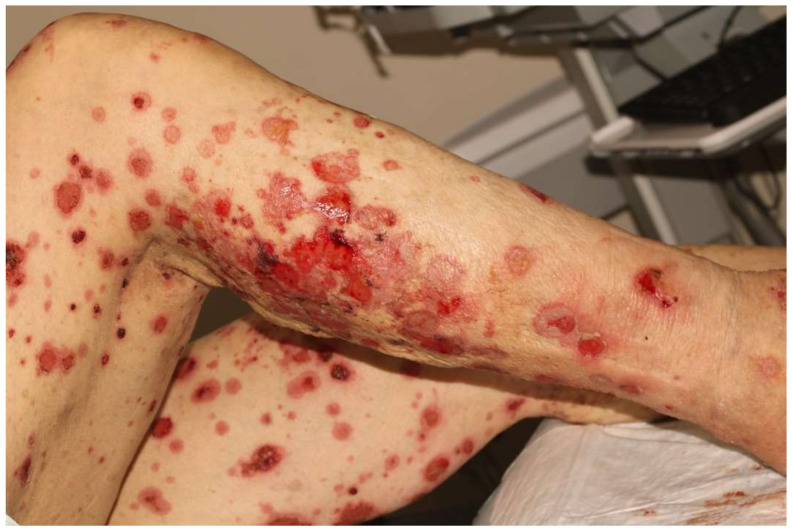
Bullous pemphigoid induced by pembrolizumab located on bilateral lower extremities.

**Figure 3 biomedicines-11-00323-f003:**
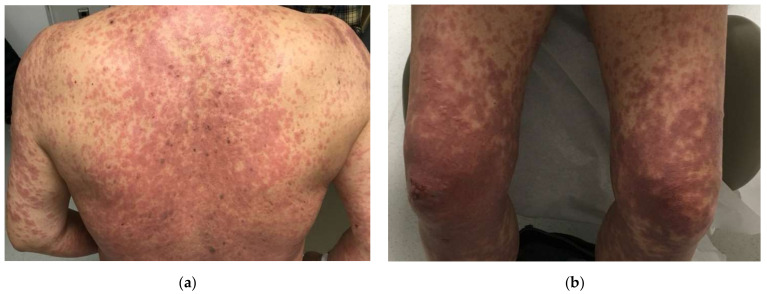
Stevens–Johnson-like reaction induced by pembrolizumab located on the (**a**) back and (**b**) bilateral lower extremities.

**Table 1 biomedicines-11-00323-t001:** Summary of diagnostic features and anti-cancer therapies associated with bullous dermatologic adverse events.

	Drugs	Clinical Features	Histopathology
**Cutaneous Bullous DAE**			
Vesiculobullous Eczema	Immune Checkpoint Inhibitors	- Prodromal erythematous plaques followed by diffuse scaly bullous eruption on the upper and lower extremities	- Spongiotic dermatitis with lymphocytic dermal infiltrate - Civatte bodies and parakeratosis - DIF negative
Hand–Foot Skin Reaction	Cytotoxic chemotherapy, targeted therapy	- Painful, sharply demarcated hyperkeratotic erythematous papules and plaques on pressure points of palmoplantar surfaces	- Vacuolar degeneration, keratinocyte apoptosis or necrosis, and intracytoplasmic eosinophilic bodies with intraepidermal cleavage - Dyskeratosis, papillomatosis, acanthosis, or parakeratosis
Bullous Toxic Erythema of Chemotherapy	Chemotherapy and targeted therapy	- Purple patches and plaques, with bulla and erosions in severe cases, favoring the hands, feet, and intertriginous skin	- Thickened epidermis with dyskeratosis and suprabasalar acantholysis as well as eccrine duct atypia
**Systemic Bullous DAE**			
Bullous Pemphigoid	Immune Checkpoint Inhibitors	- Generalized pruritus, followed by macular or tense bullae on the extremities and torso - Oral mucosal involvement in <40% of cases	- Subepidermal clefting - Eosinophils and fibrin with lymphocytes and band-like dermal infiltrate - Linear IgG/C3 at DEJ-BP180 positive±BP230
Lichen Planus Pemphigoides	Immune Checkpoint Inhibitors	- Lichenoid plaques with bullae on both plaques and unaffected skin - Mucosal involvement in about half of cases	- Subepidermal clefting with lymphocytic, eosinophilic, and neutrophilic infiltrate - DIF with IgG/C3 at DEJ - BP180 positive with distinct NC16A domain C-terminal region
Pemphigus Vulgaris	Immune Checkpoint Inhibitors	- Flaccid, nonpruritic bullae and painful erosions on the extremities, torso, and mucosal surfaces	- Suprabasal clefting with fibrin, acanthosis, eosinophils, and neutrophils in epidermis and upper dermis - DIF with IgG on keratinocytes - Desmoglein-3 positive ± desmoglein-1, and anti-desmocollin-2 and -3 - BP180/230 negative
Bullous Erythema Multiforme	Immune Checkpoint Inhibitors	- Diffuse, flaccid bullae, painful erythematous plaques, and targetoid lesions with central necrosis ± mucosal ulceration	- Blisters at DEJ with vacuolar degeneration and eosinophilic and lymphocytic infiltrate
Linear IgA Bullous Dermatosis	Immune Checkpoint Inhibitors and antimetabolite chemotherapy	- Symmetric, bullous/herpetiform lesions on the trunk and upper extremities - Bullae may coalesce into annular plaques	- Subepidermal blisters with papillary abscesses containing neutrophils ± eosinophils - DIF with linear IgA deposition ± IgG/C3 at DEJ - BP180 positive
Bullous Lupus Erythematosus	Immune Checkpoint Inhibitors	- Nonspecific prodromal rash with development of tense vesicles and bullae on the sun-exposed skin, oral mucosa, and genitals	- Subepidermal blisters with neutrophilic infiltrate at DEJ, subepidermal clefting, apoptotic keratinocytes, and basal keratinocyte vacuolization - Mucin and micro-abscesses in papillary dermis - DIF with linear IgG/C3 with granular IgM/IgA along DEJ
Stevens–Johnson Syndrome (SJS), Toxic Epidermal Necrolysis (TEN), and SJS/TEN-like Reactions	Chemotherapy, targeted therapy, and immune checkpoint inhibitors	- Prodrome of fever, sore throat, malaise, and non-pruritic truncal morbilliform rash - Progresses to maculopapular rash with blisters on dusky purpuric macules or targetoid patches and positive Nikolsky - Mucosal erosions- Followed by desquamation	- Full-thickness keratinocyte necrosis and subepidermal clefting- Sparse mononuclear dermal infiltrate and CD8+ T cells at the DEJ - Negative DIF and ELISA
Bullous Lichenoid Drug Eruption	Chemotherapy and immune checkpoint inhibitors	- Lichenoid maculopapular rash with diffuse pruritus- Develops into ill-defined tense vesicles and bullae with erosions on legs and trunk - Spares mucosa of bullae but Wickham striae may be seen	- Lichenoid interface dermatitis, focal hypergranulosis, eosinophils, and focal necrotic keratinocytes - May have focal subepidermal clefting, prominent lymphocytic infiltrate, sawtooth acanthosis, hyperkeratosis, and parakeratosis- DIF may show focal IgM - May be BP180 positive

Abbreviations: DAE = dermatologic adverse event; DIF = direct immunofluorescence; IgG/C3 = Immunoglobulin G/complement 3; DEJ = dermal epidermal junction; IgA = Immunoglobulin A; ELISA = enzyme-linked immunoassay; IgM = Immunoglobulin M.

**Table 2 biomedicines-11-00323-t002:** Bullous dermatologic adverse event Grading (Adapted from Common Terminology Criteria for Adverse Events Grading Criteria version 5) [24].

	Grade 1	Grade 2	Grade 3	Grade 4	Grade 5
**Eczema**	- Asymptomatic or mild symptoms - No intervention indicated	- Moderate - Topical or oral intervention indicated	- Severe or medically significant - IV intervention indicated	-	-
**Toxic Erythema of Chemotherapy**	- Minimal erythema, edema, or hyperkeratosis - No pain	- Blisters, peeling, fissures, bleeding - Pain - Limit instrumental ADLs	- Higher severity blisters, peeling, fissures, bleeding - Pain - Limit self-care ADLs	-	-
**Erythema Multiforme**	- BSA <10% - Asymptomatic	- BSA 10–30% - Skin tenderness	- BSA >30% BSA - Oral or genital erosions	- >30% BSA - Fluid or electrolyte abnormalities- Requires admission to ICU or burn unit	- Death
**Bullous Dermatitis, such as Hand–Foot Skin Reaction,** **Bullous Lupus Erythematosus,** **Bullous Pemphigoid,** **Bullous Lichenoid Drug Eruption, Lichen Planus Pemphigoides,** **Pemphigus Vulgaris,** **Linear IgA Bullous Dermatosis**	- Blister BSA <10% - Asymptomatic	- Blister BSA 10–30% - Blisters are painful - Limit instrumental ADLs	- Blister BSA >30% - Limit self-care ADLs	- Blister BSA >30% - Limit self-care ADLs - Fluid or electrolyte abnormalities - Requires admission to ICU or burn unit	- Death
**Stevens–Johnson syndrome(SJS) and SJS-Like Eruptions**	-	-	- Skin sloughing BSA <10% - Erythema, purpura - Epidermal and mucous membrane detachment	- Skin sloughing BSA 10–30% BSA - Erythema, purpura - Epidermal and mucous membrane detachment	- Death
**Toxic Epidermal Necrolysis**	-	-	-	- Skin sloughing BSA ≥ 30% - Erythema, purpura- Epidermal and mucous membrane detachment	- Death

Abbreviations: IV = intravenous; ADL = activities of daily living; BSA = body surface area; ICU = intensive care unit.

**Table 3 biomedicines-11-00323-t003:** Proposed pathogenesis of cutaneous and systemic bullous dermatologic adverse events (DAE).

Type of DAE	Pathogenesis	References
**Cutaneous Bullous DAE**		
Vesiculobullous eczema	- Pathogenesis related to anti-cancer therapy has not yet been postulated. - Idiopathic bullous eczema is hypothesized to result from over-expression of aquaporin 3 and -10 in keratinocytes throughout the mid and upper epidermis, resulting in epidermal fissuring and subsequent vesicle formation secondary to cutaneous water and glycerol outflow.	[23]
Hand–Foot Skin Reaction	- Caused by direct blockade of VEGFR, PDGFR, and EGFR in healthy tissue. - Eccrine excretion of inciting drug is postulated to cause direct dermal toxicity and/or inhibit receptors, leading to impaired wound healing especially in frictional areas. - Some authors have suggested that hand–foot skin reaction may be equivalent to a Koebner phenomenon, which is the development of new skin lesions secondary to trauma.	[26,27,28,29,30]
Bullous Toxic Erythema of Chemotherapy (TEC)	- Pathogenesis of cytotoxic chemotherapy therapy-induced TEC is likely related to drug accumulation in eccrine sweat glands and subsequent local toxicity. - The pathogenesis of TEC secondary to enfortumab vedotin therapy is postulated to be induced by deposition of the cytotoxic monomethyl auristatin E in tissues expressing nectin-4, such as the skin. Enfortumab vedotin induces apoptosis of keratinocytes expressing nectin-4, causing dysfunctional cell-cell adherence and bullae formation.	[31,32,33].
**Systemic Bullous DAE**		
Bullous Pemphigoid (BP)	- Activation of antibody-secreting B cells, inhibition of immunosuppressive regulatory T-lymphocytes, cross-reaction between anti-BP180 antibodies since BP180 is expressed by many tumor cells, or the triggering of clinically undetectable emerging BP by ICI.	[3,8,34,35,36,37]
Bullous Lichenoid Drug Eruption (BLDE)	- Due to exuberant lichenoid dermatitis with CD4+ and CD8+ T cell involvement. - Some have suggested that the pathogenesis of ICI-induced BLDE is similar to that of SJS/TEN, involving apoptosis of basal keratinocytes secondary to activation of CD8+ T cells by the perforin/granzyme pathway.	[38,39]
Lichen Planus Pemphigoides	- ICI may cause lichenoid dermatitis, which leads to BP180 exposure at the DEJ, allowing the host immune system to develop antibodies targeting these exposed BP180 self-antigens. This develops into epitope spreading, leading to autoimmune bullous progression of lichenoid lesions.	[40,41,42]
Pemphigus Vulgaris (PV)	- Immune-mediated T cell reaction secondary to nivolumab, triggering onset or recurrence in susceptible patients. - Spontaneous PV is thought to be the result of circulating IgG to desmoglein-3 and sometimes desmoglein-1, causing dissociation at the epidermal desmosomes and subsequent acantholysis. - It is possible that nivolumab causes an upregulation of these antibodies through a generalized increase in immune function, triggering PV.	[43]
Bullous Erythema Multiforme (BEM)	- Severe immune reaction to antigens as a result of CD4+ and CD8+ T cell imbalance, which may be caused by increased expression of Fas ligand on T cells in response to nivolumab, causing increased keratinocyte apoptosis. - Another theory involves increased differentiation of immature T cells expressing CTLA-4 in response to ipilimumab, causing a hypersensitivity loop of activated T cells to an antigen. - Similar to non-ICI induced BEM, autoreactive T cells and associated cytokines may lead to the pathologic findings of the disease state.	[44]
Linear IgA Bullous Dermatosis (LABD)	- The pathogenesis of anti-cancer drug induced LABD has not been postulated. - Spontaneous LABD involves circulating IgA anti-basement membrane zone antibodies directed against the 97 kDa portion of BP180 in the lamina lucida.	[45]
Bullous Lupus Erythematosus	- Not yet characterized.	
Stevens–Johnson Syndrome (SJS), Toxic Epidermal Necrolysis (TEN)	- Delayed-type hypersensitivity reaction in which cytotoxic T cells generate and release granulysin via the Fas/Fas ligand pathway, which leads to disseminated keratinocyte death. - PD-1, PD-L1, or CTCLA-4 inhibition leads to impaired T cell homeostasis in the skin and loss of protection from skin autoimmunity, leading to cytotoxic inflammatory reactions. - With EGFR inhibitors, it is theorized that irreversible inhibition of EGFR leads to interference of epidermal differentiation and re-epithelialization which leads to extensive erosions and the clinical appearance of SJS/TENs.	[8,39,46,47]
SJS/TEN-like Reactions	- The pathophysiology of SJS-like reactions to PD-1 inhibitors is unknown but thought to possibly be initiated by an erosive lichenoid process. The delayed nature may be due to the gradual loss of peripheral tolerance self-antigen directed T cells in the setting of rising checkpoint inhibitor concentrations over time.	[47]

Abbreviations: DAE = dermatologic adverse event; VEGFR = vascular endothelial growth factor receptor; PDGFR = platelet-derived growth factor receptor; EGFR = epidermal growth factor receptor; ICI = immune checkpoint inhibitor; DEJ = dermal epidermal junction; IgG = Immunoglobulin G; CTLA = cytotoxic T-lymphocyte–associated antigen; IgA = Immunoglobulin A; kDa = kilodalton; PD = programmed cell death protein.

## Data Availability

Data sharing not applicable.

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
