# Peer review of "A Review of Bullous Dermatologic Adverse Events Associated with Anti-Cancer Therapy"

_biomedicines, 2023, doi:10.3390/biomedicines11020323_

Round 1

Reviewer 1 Report

This is an important paper regarding bullous dermatoses related to anti-cancer therapy.  The manuscript needs some revision to be accepted for publication.

1.The term simple Bullous Dermatologic Adverse Events (BDAE) and complex BDAE are inappropriate. I think that cutaneous BDAE and systemic BDAE will be better. In Table 2, please add the subclassification column of simple and complex BDAE (rather cutaneous and systemic BDAE).

2.What is the difference between TEN/SJS and TEN/SJS-like eruption? Please define.

3.The order of tables should be changed; Table 2 should be the first.

4.Schemes for the pathogenesis of typical BDAE should be presented; immune-check-point inhibitors, anti-EGFR antibody, anti-receptor kinase inhibitors, toll-like receptor agonists, etc.

Author Response

01/18/2023

Dear Prof. Dr. Shaker A. Mousa,

We are pleased to submit the revision of our manuscript, “A Review of Bullous Dermatologic Adverse Events Associated with Anti-Cancer Therapy,” an invited review article for Biomedicines’s special issue “Hypersensitivity to Drugs and Vaccines: Molecular Basis and Translational Research.” In this letter we summarize our responses to the issues raised by the reviewers.

Reviewer 1:

Comment #1: The term simple Bullous Dermatologic Adverse Events (BDAE) and complex BDAE are inappropriate. I think that cutaneous BDAE and systemic BDAE will be better. In Table 2, please add the subclassification column of simple and complex BDAE (rather cutaneous and systemic BDAE).

We appreciate the reviewer’s feedback. The terminology was adjusted to cutaneous and systemic bullous DAE, as recommended. In Table 1, subclassification rows were added, and the table was reordered appropriately. We thank the reviewer for this suggestion.

Comment #2: What is the difference between TEN/SJS and TEN/SJS-like eruption? Please define.

In lines 515-517 we have added a sentence to address the difference. Thank you for this suggestion.

Comment #3: The order of tables should be changed; Table 2 should be the first.

Tables 1 and 2 were switched as recommended.

Comment #4: Schemes for the pathogenesis of typical BDAE should be presented; immune-check-point inhibitors, anti-EGFR antibody, anti-receptor kinase inhibitors, toll-like receptor agonists, etc.

We have added Table 3 to summarize the postulated pathogenesis of each of the BDAE based on the information included in the body of the manuscript.  Thank you for this recommendation.

Thank you for the opportunity to revise and resubmit our manuscript. Please let me know if you have any questions or need further clarification.

Sincerely,

Alina Markova, MD

Director of Inpatient Consultative Dermatology

Dermatology Service

Department of Medicine

Memorial Sloan Kettering Cancer Center

545 E 73rd Street

New York, New York 10021

markovaa@mskcc.org

Roni P Dodiuk-Gad, MD

Division of Dermatology

Department of Medicine

University of Toronto, Toronto, ON M5S 1A1, Canada

Emek Medical Center, Afula, Israel.

Bruce Rappaport Faculty of Medicine, Technion Institute of Technology, Haifa 3525433, Israel

Reviewer 2 Report

The article “A Review of Bullous Dermatologic Adverse Events Associated with Anti-Cancer Therapy” is very interesting and I have some comments with intention to improve the article.

-        Introduction: Indicate the most appropriate reference for the phrase on line 39-44.” Bullous DAEs consist of vesiculobullous eczema, hand foot skin reaction, toxic erythema of chemotherapy, bullous pemphigoid, bullous lichenoid drug eruption, lichen planus pemphigoides, pemphigus vulgaris, bullous erythema multiforme, linear IgA bullous dermatosis, bullous lupus erythematosus (Stevens-Johnson Syndrome SJS)/toxic epidermal necrolysis and SJS-like eruptions, and non-specific bullous drug eruption”

-        Where do the figures come from?

-        Based on which criteria you selected the bullous diseases that have stood out

-        Indicate the first time in the text that the meaning of the abbreviations (line 132, BP; line 145 VEGF…..

-        Some therapeutic recommendation should be included, for example what type of topical corticosteroid therapy, if a high potency corticosteroid, medium potency... Or in the case of systemic corticosteroids, what guidelines are proposed per kg of weight of the patient and what type of corticosteroid depending on the cancer therapy

-        References: Adapt the references to the conditions of the journal.

Author Response

01/18/2023

Dear Prof. Dr. Shaker A. Mousa,

We are pleased to submit the revision of our manuscript, “A Review of Bullous Dermatologic Adverse Events Associated with Anti-Cancer Therapy,” an invited review article for Biomedicines’s special issue “Hypersensitivity to Drugs and Vaccines: Molecular Basis and Translational Research.” In this letter we summarize our responses to the issues raised by the reviewers.

Reviewer 2:

Comment #1: The article “A Review of Bullous Dermatologic Adverse Events Associated with Anti-Cancer Therapy” is very interesting and I have some comments with intention to improve the article.

We thank the reviewer for their kind words and suggestions.

Comment #2: Introduction: Indicate the most appropriate reference for the phrase on line 39-44.” Bullous DAEs consist of vesiculobullous eczema, hand foot skin reaction, toxic erythema of chemotherapy, bullous pemphigoid, bullous lichenoid drug eruption, lichen planus pemphigoides, pemphigus vulgaris, bullous erythema multiforme, linear IgA bullous dermatosis, bullous lupus erythematosus (Stevens-Johnson Syndrome SJS)/toxic epidermal necrolysis and SJS-like eruptions, and non-specific bullous drug eruption”

We have added appropriate citations to the end of this sentence. We thank the reviewer for this suggestion.

Comment #3: Where do the figures come from?

The patients were seen at Memorial Sloan Kettering Cancer Center, New York, New York.

Comment #4: Based on which criteria you selected the bullous diseases that have stood out.

In lines 125-126 we added a sentence to the methods section to address the reviewer’s comment: “Bullous DAE included were selected based on clinical presentation and relevance as determined by leading experts in the field.” We thank the reviewer for this suggestion.

Comment #5: Indicate the first time in the text that the meaning of the abbreviations (line 132, BP; line 145 VEGF)

We have made these corrections in line 146 and 165. We thank the reviewer for catching this.

Comment #6: Some therapeutic recommendation should be included, for example what type of topical corticosteroid therapy, if a high potency corticosteroid, medium potency... Or in the case of systemic corticosteroids, what guidelines are proposed per kg of weight of the patient and what type of corticosteroid depending on the cancer therapy.

Thank you for this suggestion. A large majority of the included articles did not specify the potency and dosage of therapy. Given this, we feel there is not enough consistent literature to allow for us to make accurate clinical recommendations in this regard.

Comment #7: References: Adapt the references to the conditions of the journal.

References have been updated using the Biomedicines/MDPI EndNote format. Thank you.

Thank you for the opportunity to revise and resubmit our manuscript. Please let me know if you have any questions or need further clarification.

Sincerely,

Alina Markova, MD

Director of Inpatient Consultative Dermatology

Dermatology Service

Department of Medicine

Memorial Sloan Kettering Cancer Center

545 E 73rd Street

New York, New York 10021

markovaa@mskcc.org

Roni P Dodiuk-Gad, MD

Division of Dermatology

Department of Medicine

University of Toronto, Toronto, ON M5S 1A1, Canada

Emek Medical Center, Afula, Israel.

Bruce Rappaport Faculty of Medicine, Technion Institute of Technology, Haifa 3525433, Israel

Round 2

Reviewer 1 Report

The authors well addressed the issues I pointed out and appropriately revised the article.